# Diagnostic Utility of pH-MII Monitoring in Preschool Children with Recurrent Wheeze and Suspected Gastroesophageal Reflux Disease: A Prospective Study

**DOI:** 10.3390/diagnostics13233567

**Published:** 2023-11-29

**Authors:** Ivan Pavić, Roberta Šarkanji-Golub, Iva Hojsak

**Affiliations:** 1Department of Pulmonology, Immunology, Rheumatology and Allergology, Children’s Hospital Zagreb, 10000 Zagreb, Croatia; 2School of Medicine, University of Split, 21000 Split, Croatia; 3Department of Cytology, Children’s Hospital Zagreb, 10000 Zagreb, Croatia; roberta.sago@gmail.com; 4Referral Center for Pediatric Gastroenterology and Nutrition, Children’s Hospital Zagreb, 10000 Zagreb, Croatia; ivahojsak@gmail.com; 5School of Medicine, University of Zagreb, 10000 Zagreb, Croatia; 6School of Medicine Osijek, University Josip Juraj Strossmayer, 31000 Osijek, Croatia

**Keywords:** recurrent wheeze, preschool children, gastroesophageal reflux, pH-multichannel intraluminal impedance monitoring

## Abstract

Background: Recurrent wheezing and gastroesophageal reflux disease (GERD) are common in young children, with a suggested but challenging link between them. This study aimed to investigate the diagnostic value of pH-MII monitoring in preschool children with recurrent wheezing and evaluate GERD-related therapy effects. Methods: Children under 6 years with recurrent wheeze were eligible. The pH-MII monitoring was conducted in those clinically suspected of GERD’s involvement. Flexible bronchoscopy with bronchoalveolar lavage (BAL) was performed in severe cases. The primary outcome was the difference in wheezing episodes between proven GERD and non-GERD groups. Secondary outcomes included GERD therapy impact and predictive factors for wheezing reduction. Results: Of 66 children (mean age 3.9 years), 71% had proven GERD on pH-MII. Compared to the non-GERD group, the GERD group had higher total, liquid, mixed, and gas reflux episodes, as well as more acidic and weakly acidic episodes. GERD treatment significantly reduced wheezing episodes. PPI (proton pump inhibitor) introduction was associated with ≥50% wheezing reduction. Children with GERD showed ≥50% wheezing reduction more frequently than those without GERD. PPI usage, higher total GER episodes, acidic episodes, and liquid and proximal episodes on MII predicted ≥50% wheezing reduction. No significant BAL differences were observed between GERD and non-GERD groups. Conclusions: The pH-MII monitoring is valuable in diagnosing GERD-related wheezing in preschool children. GERD therapy, particularly PPI usage, was associated with reduced wheezing episodes. The pH-MII parameters correlated with wheezing reduction, suggesting their potential predictive role. BAL did not differentiate between GERD and non-GERD cases.

## 1. Introduction

Recurrent wheezing and gastroesophageal reflux (GER) are common conditions in infants and young children. For many years, gastroesophageal reflux disease (GERD) has been suggested as one of the several causes of wheezing in preschoolers, although it is difficult to prove the link between these two conditions. Previously, pH-metry alone was used to show that a high proportion of preschoolers with recurrent wheezing had GERD [1,2,3,4]. However, pH-metry is blind to weakly acidic reflux (WAR). This could be an important limitation as the advent of multichannel intraluminal impedance (pH-MII) has shown that not only acid reflux (AR), but also weakly acidic reflux (WAR), which cannot be detected by pH-metry alone, can be a major contributor to reflux-related respiratory disease [5,6,7]. In addition, the pH-MII testing allows the determination of the composition (liquid, gas, and mixture), the level reached by the refluxate, the differentiation between swallowing and reflux episodes, and the measurement of symptoms associated with reflux [8]. This could be of particular interest in the assessment of extraesophageal manifestations of GERD, especially in younger children, as the gastric contents are buffered by the frequent feedings and the composition of the feedings differs from those of older children, which could limit the usefulness of pH-metry alone in the assessment of reflux-related respiratory manifestations. Data on pH-MII monitoring in preschool children with recurrent wheeze are lacking. In addition, much less is known about the prevalence of WAR in preschool children with recurrent wheeze. Therefore, the clinical relevance of the temporal association of preschool wheeze and GERD should be further investigated.

The aim of this prospective study is to investigate the diagnostic utility of pH-MII monitoring in preschool children with recurrent wheeze.

## 2. Materials and Methods

### 2.1. Study Design

All children younger than 6 years who were referred to the Department of Pediatric

Pulmonology, Children’s Hospital Zagreb (Croatia), for further investigation of recurrent wheeze between June 2018 and July 2022 were eligible for inclusion. Recurrent wheezing is defined as four or more episodes in the previous year [1].

Inclusion criteria were: children younger than 6 years of age with four or more episodes of wheezing in the previous 12 months before study enrolment, with no proven etiology, and who had a possible association with GERD as one of the possible causes of their unexplained respiratory symptoms. All included children had no other gastrointestinal symptoms suggestive of GERD. Children with current respiratory infections, swallowing disorders, airway and other structural abnormalities, upper airway cough syndrome, suspected and proven asthma, and heart disease, as well as known antibiotic therapy in the month before admission, were excluded from the study.

In children with recurrent wheezing, 24 h pH-MII monitoring was performed because GERD and/or aspiration were clinically suspected as one of the possible causes. In addition, flexible bronchoscopy with bronchoalveolar lavage (BAL) was performed in children with more severe clinical symptoms. Children with more severe symptoms include those with a higher number of wheezing episodes, those who required hospitalisation, those who were slow to recover from wheezing episodes and those who were refractory to first-line therapy. The indication for the diagnostic procedures was given by the same pediatric pulmonologists at the tertiary medical center.

Before the enrolment into the study, the indication and the procedure were explained and discussed with the child’s parents and caregivers. Only children who met the inclusion criteria and whose parents/caregivers signed an informed consent form were included in the study. Children whose symptoms could not be explained by another condition and who underwent further investigations to rule out other conditions prior to admission received a detailed medical history, a thorough physical examination by a single pediatric pulmonologist, a formal ear, nose, and throat (ENT) examination including a laryngoscopic examination, a pulmonary function test (for those who could co-operate), a chest X-ray, blood tests (complete blood count, differential white blood cell count, C-reactive protein, immunoglobulins and IgG subclasses, and total IgE), skin prick tests for common allergens, and sweat chloride testing.

The study was conducted in accordance with the Declaration of Helsinki and approved by the Ethics Committee of the Children’s Hospital Zagreb.2.2. Esophageal pH-MII Monitoring Procedure

All children underwent 24 h pH-MII monitoring with an ambulatory pH-MII system (Ohmega, MMS, Enschede, The Netherlands).

The combined MII-pH monitoring was recorded with a catheter consisting of an antimony pH sensor and six pairs of impedance electrodes for measuring intraluminal impedance. Age-appropriate catheters were used (according to length/height): infant and paediatric catheters. At the beginning of the procedure, the pH electrode was calibrated with pH 4.0 and 7.0 buffer solutions. Each catheter was used only once and none of the catheters malfunctioned. The catheter was inserted nasally and positioned so that the pH electrode was on the third vertebral body above the diaphragmatic angle. The correct position of the catheter was confirmed by X-ray. Parents were trained to keep a careful diary and to use an event marker in the data logger to record mealtimes, body position, daily activities, and timing of laryngopharyngeal symptoms. The MII-pH recordings were uploaded to a personal computer at the end of the recording period and manually analyzed using the MMS software package (Version 9.6a).

Impedance recordings were analyzed using the criteria described in a consensus statement on indications, methodology, and interpretation of combined pH-MII monitoring in children [9,10,11].

Gastroesophageal reflux disease (GERD) was defined on pH-metry as a reflux index (RI) >10% in infants and >7% in older children [9,10,11]. Gastroesophageal reflux detected by MII was categorised as acidic when the pH drops from above to below 4.0, weakly acidic when the pH is between 4.0 and 7.0, and weakly alkaline (non-acidic) when the intraesophageal pH remains above 7.0 during an episode of reflux detected by MII. The MII results were defined as abnormal if the measurement met two of three of the following criteria: symptom index (SI) ≥ 50%, symptom sensitivity index (SSI) ≥ 10%, and symptom association probability (SAP) ≥ 95% [11]. Patients were followed up for 12 months.

### 2.2. Bronchoscopy and Bronchoalveolar Lavage

Flexible bronchoscopy with BAL was performed according to the European Respiratory Society guidelines [12]. Bronchoscopy and BAL were performed by the same multidisciplinary team of pediatric pulmonologist and anesthesiologists and were performed under general anesthesia. All procedures were performed in the operating room.

BAL was performed in the right middle lobe or selected at the time of bronchoscopy depending on the findings. Total cell count was determined in a Füchs–Rosenthal counting chamber. Smears for cell differentiation were prepared by cytocentrifugation. Cell differentiation was performed by microscopy on cytospin slides after staining with May–Grünwald–Giemsa stain. Cell evaluation was performed by the same cytologist, who was blinded to the bronchoscopy result, BAL culture, and clinical findings of the patients, using a light microscope. Oil Red O (ORO) staining was used to detect LLM, count them, and express them as a ratio to the total amount of alveolar macrophages (AM) (ratio (%) LLAM/AM), as previously described [13]. For microbiological studies, the first non-centrifuged aliquots of BAL were used. Bronchoscopy and BAL were well tolerated by all subjects.

### 2.3. Outcomes

The primary outcome was to determine the difference in wheezing episodes in children with proven GERD compared to children with normal GERD based on MII. Secondary outcomes were to assess the effect on the GERD therapy on the wheezing episodes (12 and 6 months before and 12 and 6 months after the treatment), to assess pH-MII predictive factors for the significant decrease in wheezing episodes in the period of the 6 months before and 6 months after pH-MII study. A significant decrease was defined as at least 50% reduction in the number of episodes.

In addition, the difference between BAL findings in children with GERD and children without GERD was investigated in children who underwent bronchoscopy with bronchoalveolar lavage (BAL) at the same time as pH-MII monitoring. The correlation between BAL findings and 24 h pH-MII was determined to assess the difference between children with GERD and children without GERD.

### 2.4. Statistics

Shapiro–Wilk test determined that data were not normally distributed. The differences between categorical variables were assessed by a chi-square test. The differences for non-categorical data were assessed by a non-parametric Mann–Whitney U-test. The *p* values less than 0.05 were considered significant. Correlation was assessed by Pearson or Spearman correlation coefficient. Binary logistic regression was used to determine prognostic factors for significant decrease in the number of wheezing episodes defined as 50% reduction in the wheezing episodes. First univariate logistic regression was performed followed by multivariate logistic regression (forward stepwise). Statistical analysis was performed using SPSS 26.0 (Chicago, IL, USA) statistical software.

## 3. Results

There were altogether 66 children included in the study, 22 (33%) female, age 3.9 (0.2 SD) years. GERD was found in 46 (71%) based on MII and only in 8 (12%) based on pH-metry alone.

The difference in demographic data between children with GERD vs. those without GERD defined by MII is presented in Table 1.

Table 2 presents difference in the pH-MII characteristics between groups. As expected, children with GERD had a significantly higher number of total GER episodes, liquid, mixed, and gas episodes. Furthermore, children with GERD had more acidic and weakly acidic episodes.

Children with GERD had more wheezing episodes 12 months prior to the pH-MII (Table 3). After the pH-MII and after therapy was introduced, children with GERD had significantly higher reduction in the wheezing episodes (Table 3).

As expected, children with GERD received more GERD-related therapy, especially PPI (Table 4).

In 46 children with proven GERD, a PPI inhibitor was introduced in 31 (67%) children, 26 (57%) received PPI alone, and 5 (11%) in combination with alginate. Alginate alone was used in 9 (20%) children with proven GERD and 5 (11%) in combination with PPI. In other patients (n = 6, 13%) only symptomatic measures were recommended and 37 (80%) in combination with other therapy.

Out of the whole cohort, in 65 (98%) children, the number of wheezing episodes decreased after pH-MII; the median number of wheezing episodes decreased from 3 (2–6) in the period of 6 months before pH-MII to 0.5 (0.3) in the 6 months period after pH-MII (*p* < 0.001).

Overall, at least 50% decrease in the number of wheezing episodes was seen in 60 (91%) children; in all children (*n* = 46, 100%) with GERD vs. 14 (70%) in children without GERD (*p* < 0.001) (Table 3). While ≥75% decrease in wheezing episodes was seen in 16 children who were all patients with GERD (*p* = 0.002). None of the children without GERD had ≥ 75% improvement in the number of wheezing episodes.

A 50% decrease in the number of wheezing episodes was significantly associated with PPI use (coef. 0.326, *p* = 0.008) but not with alginate use (coef. 0.179, *p* = 0.151). All children in whom PPI was introduced had 50% of reduction in wheezing.

A univariate logistic regression identified factors associated with ≥50% of reduction in wheezing episodes (Figure 1).

Odds ratio for ≥50% of reduction in wheezing episodes was higher when the number of total GER episodes, acidic episodes, and liquid and proximal episodes on MII increased. Multivariate analysis (forward stepwise) included the above-mentioned factors together with sex and age, and identified only proximal episodes as significant (hazard ratio (HR) 1.3; 95% CI 1.03–1.62; *p* = 0.026).

Bronchoscopy was performed in 37 (56%) of children. Macroscopic changes were found in 18 (49%) children; 14 (50) in GERD group vs. 4 (44%) in no-GERD group (*p* = 0.772).

Difference in bronchoalveolar lavage between patients with GERD vs. those without GERD on MII are presented in Table 5. There was no statistically significant difference in the percentage of cells between those two groups. There was no significant correlation between lipid-laden macrophages and reflux index, total number of GER, acidic, weakly acidic and non-acidic GER, liquid GER, and proximal GER episodes based on MII.

Out of 37 patients, BAL was positive for microbiology in 8 children (22%).

## 4. Discussion

Recurrent wheezing in preschool children is a common clinical challenge, and its association with GERD has long been suspected. This prospective study sought to elucidate the diagnostic utility of pH-MII monitoring in preschool children with recurrent wheeze, focusing on the relationship between GERD and respiratory symptoms. The study’s findings shed light on the significance of pH-MII monitoring in diagnosing and managing GERD-related wheezing episodes in this vulnerable population.

The results of our study showed that a high proportion of children under 6 years of age with recurrent wheezing had GERD, which was mostly diagnosed by the combined MII method than by the pH-metry alone. This confirms that not only AR, but also WAR can cause GERD-related wheezing in this age group of children. In addition, children with GERD had more wheezing episodes 12 months before pH-MII. Moreover, children with GERD had a significantly higher decrease in wheezing episodes after pH-MII and after the introduction of anti-reflux therapy.

Our findings underscore the diagnostic utility of pH-MII monitoring in preschool children with recurrent wheezing, revealing a significantly higher proportion of GERD cases compared to pH-metry alone. This aligns with recent studies that have emphasized the limitations of pH-metry in detecting WAR [14,15,16,17], further highlighting the importance of pH-MII in providing a more accurate representation of reflux events and their associations with clinical symptoms. Notably, we observed that GERD was diagnosed in 71% of cases using pH-MII compared to only 12% with pH-metry alone. This confirms the significance of including WAR assessment in reflux-related studies, particularly in the context of respiratory manifestations.

Previously, Rosen and colleagues found a high rate of abnormal reflux testing in their study in which 112 patients aged 1 to 18 years underwent combined bronchoscopy, laryngoscopy, endoscopy, and a pH-MII monitoring to assess cough and wheeze; 58% of patients were diagnosed with an abnormality based on pH-MII or endoscopy [18]. This is lower than the percentage found in our study, 71%; but, interestingly, we found a higher percentage of WAR. The main reason for this discrepancy could be differences in the age of the study participants, as we studied children up to 6 years of age. It is known that in younger children the stomach contents are buffered by the frequent feedings, as well as the known differences in the composition and volume of the feedings compared to those of older children.

It is important to note that not all individuals with reflux experience respiratory symptoms, and the relationship between reflux and wheezing can vary widely. Factors such as the type and amount of refluxed material, the sensitivity of the airway lining, individual differences in neural responses, and underlying respiratory conditions can all influence the likelihood and severity of reflux-related wheeze [19].

Previously, several studies measuring pH with one or two probes were conducted to investigate a relationship between preschool wheezing and GERD, with varying results [2,3,4,20]. In addition to the differences in the methodology used in the different studies, other possible reasons for the discrepancies could be the different sample sizes and clinical characteristics of the children included in the studies. Apparently, previous studies did not evaluate all patients uniformly and relied on pH monitoring, which cannot detect WAR. This is particularly problematic as a larger number of pediatric refluxes are non-acidic (WAR or weakly alkaline) [21,22,23,24,25].

Zenzeri and colleagues in their prospective, controlled study showed that GERD-related respiratory symptoms in children older than 1 year are significantly related to non-acid reflux than in children with GERD-related gastrointestinal symptoms, supporting the hypothesis that reflux-related respiratory symptoms are less related to acidity than gastrointestinal symptoms [26]. This is in line with the results of our study, emphasizing the importance of the pH-MII monitoring in the diagnostic workup of children with recurrent wheezing episodes highly suspected that are induced by GER.

Our study contributes to the growing body of evidence suggesting that both AR and WAR are implicated in the development of GERD-related wheezing in preschool children. The identification of WAR as a significant contributor to reflux-related respiratory disease is particularly noteworthy, as its detection was previously hindered by pH-metry’s limitations in capturing weakly acidic episodes. This finding calls for a paradigm shift in understanding the mechanisms underlying reflux-induced wheezing in this vulnerable population.

The clinical implications of our study are two-fold. First, the accurate diagnosis of GERD through pH-MII monitoring enables more targeted therapeutic interventions. We observed that anti-reflux therapy led to a significant reduction in wheezing episodes, further reinforcing the notion that GERD plays a crucial role in driving respiratory symptoms in this age group. This positive treatment response underscores the importance of considering GERD as a potential contributor to recurrent wheezing and emphasizes the need for early intervention to improve the quality of life for affected children and their families. The marked reduction in wheezing episodes following the introduction of GERD-related therapy, particularly proton pump inhibitors (PPIs), adds weight to the argument that addressing GERD can lead to tangible clinical improvements.

Second, our findings highlight the intricate interplay between GERD and wheezing in preschool children. While the mechanisms by which GERD triggers respiratory symptoms are not yet fully understood, these results support the notion that reflux-induced microaspiration or other direct effects on airway receptors could be involved. The study also raises the question of whether early intervention and aggressive treatment of GERD in preschool children might have the potential to alter the natural history of respiratory conditions, possibly preventing the progression to more severe or persistent wheezing disorders. The identification of GERD as a significant trigger for wheezing episodes further reinforces the importance of a holistic approach to patient management, considering both respiratory and gastroesophageal factors.

Moreover, the study’s exploration of the association between specific reflux patterns and recurrent wheezing provides valuable insights into the potential mechanisms underlying GERD-related respiratory symptoms. Children with proven GERD exhibited a higher frequency of total GER episodes, including liquid, mixed, and gas reflux episodes, along with increased acidic and weakly acidic reflux events. This observation not only reinforces the hypothesis that reflux, irrespective of its acidity, contributes to respiratory symptoms but also underscores the importance of personalized approaches to treatment. The ability of pH-MII to differentiate between different reflux patterns offers clinicians a nuanced view of each patient’s reflux profile, enabling more targeted therapeutic strategies.

Research is ongoing to further understand the precise mechanisms underlying reflux-related wheeze, as well as to identify potential biomarkers and therapeutic targets [27]. Additionally, the use of advanced diagnostic techniques like pH-MII has provided researchers and clinicians with a better understanding of the types of refluxes (acidic, weakly acidic, and non-acidic) that contribute to respiratory symptoms, which can help tailor treatment strategies more effectively.

Intriguingly, the relationship between GERD and airway inflammation, as assessed through bronchoscopy and bronchoalveolar lavage (BAL), revealed no significant differences in the BAL findings between children with GERD and those without. This suggests that GERD-related respiratory symptoms might not always be accompanied by detectable airway inflammation, which is in line with previous studies [28,29,30]. While this finding may challenge the conventional understanding of reflux-related airway involvement, it underscores the complexity of the relationship between GERD and respiratory manifestations.

The predictive factors identified from pH-MII monitoring that are associated with a significant reduction in wheezing episodes provide an additional layer of clinical utility to this study. The ability to identify children who are more likely to benefit from GERD therapy based on their reflux patterns could lead to a more tailored and efficient approach to management. This predictive aspect of pH-MII monitoring aligns with the current trend in medicine towards personalized treatment strategies and the optimization of therapeutic interventions.

The study’s comprehensive design aimed to address several critical aspects of the relationship between GERD and recurrent wheezing in preschool children. By employing a prospective approach and incorporating pH-MII monitoring, the study sought to overcome some of the limitations of prior research, particularly those related to the detection of WAR reflux episodes. The inclusion of both AR and WAR monitoring is pivotal, as it better represents the full spectrum of GERD-related events that could contribute to respiratory symptoms. In addition to the broad insights gained from our study, the multivariate analysis revealed that proximal episodes were the sole significant factor associated with a marked reduction in wheezing episodes. This specific reflux pattern emerged as a key predictor of therapeutic response to GERD therapy in our cohort. This result underscores the importance of considering not only the presence of GERD but also the nature and location of reflux events when tailoring interventions for preschool children with recurrent wheeze. Further investigation into the clinical implications of proximal episodes and their role in GERD-related respiratory symptoms may provide additional insights for personalized management strategies.

Despite the valuable insights gained from this study, certain limitations warrant consideration. The relatively small sample size and single-center nature of the study may limit the generalizability of the findings to the broader population. The lack of a control group without recurrent wheeze could affect the robustness of the findings. Additionally, while pH-MII offers a more comprehensive assessment of reflux events, its limitations, such as its invasive nature and potential discomfort for young children, should be acknowledged. We recognise that the use of a control pH-MII monitor would be better for outcome measurements, but due to its invasive nature and potential discomfort for young children, we have decided not to use it. Moreover, while the study highlights the diagnostic and therapeutic benefits of pH-MII monitoring, it does not address potential complications, cost-effectiveness, or long-term outcomes associated with GERD therapy in this population. Nonetheless, the results point up the importance of pH-MII monitoring as an invaluable tool for assessing GERD-related wheezing.

In conclusion, this prospective study underscores the significance of pH-MII monitoring in diagnosing and managing GERD-related recurrent wheezing in preschool children. The high prevalence of GERD identified through pH-MII emphasizes the importance of detecting WAR episodes, which are often missed by traditional pH-metry. The study’s findings provide compelling evidence for the clinical benefit of GERD therapy, particularly PPIs, in reducing wheezing episodes in this population. The identification of predictive factors from pH-MII results further contributes to the personalized management of these children. While the relationship between GERD and airway inflammation warrants further exploration, this study significantly advances our understanding of GERD’s role in preschool wheezing and offers valuable insights for clinical practice. Further research with larger cohorts and longitudinal follow-up is warranted to validate these findings and to uncover the long-term outcomes of GERD-related therapy in preschool children with recurrent wheeze.

## Figures and Tables

**Figure 1 diagnostics-13-03567-f001:**
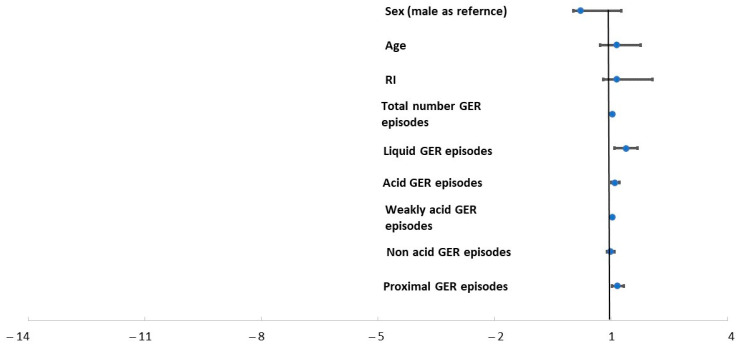
Univariate binary logistic regression showing possible predictive factors for 50% reduction in wheezing episodes.

**Table 1 diagnostics-13-03567-t001:** Difference in demographic data between patients with GERD vs. those without GERD based on MII.

	GERD (*n* = 46)	No GERD (*n* = 20)	*p*
Age, years (median, range)	5.1 (2.1–6)	5.5 (2.3–6.1)	0.214
Sex (female, %)	11 (24%)	11 (55%)	0.014
Body weight for age Z score (median, range)	−0.7 (−3 to 2.7)	0.08 (−2.3 to 1.6)	0.345
Body height for age Z score (median, range)	1.4 (−1.7 to 3.9)	1.1 (−0.2 to 3.4)	0.724
BMI for age Z score (median, range)	0.58 (−0.9 to 3.5)	0.56 (−1.5 to 2.5)	0.212

**Table 2 diagnostics-13-03567-t002:** Difference in the pH-multichannel intraluminal impedance monitoring characteristics between patients with GERD vs. those without GERD based on MII expressed as median and range.

	GERD (*n* = 46)	No GERD (*n* = 20)	*p*
Number of reflux episodes on pH metry	45 (1–144)	19.5 (6–64)	0.066
Reflux index	1.3 (0.2–3.9)	2.5 (0–24.5)	0.225
Number of reflux episodes on MII	125 (64–512)	7.6 (22–195)	<0.001
Liquid reflux episodes	19 (1–94)	12 (0–31)	0.017
Mixed reflux episodes	34 (5–110)	9.5 (1–31)	<0.001
Gas reflux episodes	65 (4–468)	44 (16–157)	0.046
Acid reflux episodes on MII	32 (1–38)	10 (0–99)	0.004
Weakly acidic reflux episodes on MII	91 (32–453)	57 (21–178)	0.001
Non-acid reflux episodes on MII	2 (0–143)	2.5 (0–40)	0.482
Proximal reflux episodes on MII	43 (17–118)	23.5 (0–55)	<0.001
Bolus clearance time	9.2 (6.1–19.2)	9.35 (6–15.9)	0.908

**Table 3 diagnostics-13-03567-t003:** Difference in number of wheezing episodes before and after the pH-multichannel intraluminal impedance monitoring between patients with GERD vs. those without GERD based on MII expressed as median and range.

	GERD (*n* = 46)	No GERD (*n* = 20)	*p*
Number of wheezing episodes 6 months before pH-MII	3 (2–6)	3 (2–4)	0.16
Number of wheezing episodes 12 months before pH-MII	6 (4–14)	5 (4–8)	0.009
Number of wheezing episodes 6 months after pH-MII	0 (0–2)	1 (0–3)	0.105
Number of wheezing episodes 12 months after pH-MII	0 (0–4)	1 (0–5)	0.395
The difference in the number of episodes after vs. before pH-MII	3 (1–4)	2 (0–4)	0.005
Percentage of reduction in wheezing episodes before vs. after pH-MII (%)	100 (50–100)	67 (0–100)	0.038

**Table 4 diagnostics-13-03567-t004:** Difference in the antireflux therapy (AR) between patients with GERD vs. those without GERD based on MII.

	GERD (*n* = 46)	No GERD (*n* = 20)	*p*
AR therapy before pH-MII	7 (15%)	1 (5%)	0.242
Symptomatic measures before pH-MII	4 (9%)	1 (5%)	0.602
AR therapy after pH-MII	43 (93%)	6 (30%)	<0.001
PPI after pH-MII	31 (67%)	3 (15%)	<0.001
Alginate after pH-MII	14 (30%)	2 (10%)	0.075
Symptomatic measures after pH-MII	43 (93%)	6 (30%)	<0.001

**Table 5 diagnostics-13-03567-t005:** Difference in bronchoalveolar lavage between patients with GERD vs. those without GERD based on MII.

	GERD (*n* = 28)	No GERD (*n* = 9)	*p*
Number of cells/μL (median, range)	108 (34–6055)	146 (78–4056)	0.475
Macrophage, % (median, range)	91 (13–99)	91 (37–96)	0.931
Lipid-laden macrophages, % (median, range)	12 (0–38)	1 (0–24)	0.144
Lymphocyte, % (median, range)	3 (0–8)	4 (2–6)	0.062
Monocyte, % (median, range)	1 (0–2)	0 (0–2)	0.821
Neutrophils, % (median, range)	3 (0–82)	3 (0–60)	0.715
Eosinophils, % (median, range)	0.5 (0–6)	1 (0–2)	0.566

## Data Availability

The data presented in this study are available on request from the corresponding author. The data are not publicly available due to privacy concerns.

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
