# Peer review of "Diagnostic Utility of pH-MII Monitoring in Preschool Children with Recurrent Wheeze and Suspected Gastroesophageal Reflux Disease: A Prospective Study"

_diagnostics, 2023, doi:10.3390/diagnostics13233567_

Round 1

Reviewer 1 Report

Comments and Suggestions for Authors

The authors present an interesting study that aims to determinethe diagnostic value of pH-MII monitoring in preschool children with recurrent wheezing and evaluate GERD-related therapy effects. The author found that 66 children, 71% had proven GERD on pH-MII.

There are several major concerns that needs to be addressed;

1. The justification of this research seems inadequate. Why GERD needs to be assessed in children with wheezing

2. How wheeze was diagnosed? How do authors exclude other concurrent pathology with wheeze? During the bronchoscopy, how was the presence of LPR-related findings excluded? Where ORL team involved?

3. Justification of PPI in these children. PPT could exacerbate respiratory symptoms. Please clarify this

4. Outcome measurement in this study is poorly conducted.

Reviewer 2 Report

Comments and Suggestions for Authors

I would like to congratulate the authors of the interesting article titled “Comprehensive assessment of gastroesophageal reflux in preschool children with recurrent wheeze: role of pH-multichannel intraluminal impedance monitoring and therapeutic implications.”

Title should include the type of the study. The abstract gives sufficient information about the study. In the introduction, the authors provide sufficient background for conducting research, including the current state of knowledge and knowledge gaps, and formulate research objectives. The material and methods section is written very carefully and would allow the research to be repeated by other scientists. Results are clear. Tables should include information on which type of measures of central tendency (median or mean) and dispersion (standard deviation, range, interquartile range) were used. The tables are quite difficult to read. The authors could consider using graphs for the most important results. The discussion in a sufficiently in-depth way relates the results obtained by the authors to the current literature and describes future research directions.

Overall, the article is interesting and may be an additional source of knowledge for doctors dealing with wheezing and reflux in children.

Comments on the Quality of English Language

Minor editing required

Reviewer 3 Report

Comments and Suggestions for Authors

Recommendations for authors

Please specify what Z1-Z3 and bolus CT means (from Table 2.)

Please specify which are the severe symptoms for patients who underwent BAL.

Also describe the pH-metry procedure.

The text on lines 193-196 is confusing. (In 46 children who had proven GERD PPI inhibitor was introduced in 31 (67%) children, 26 (57%) received only PPI and 5 (11%) in combination with alginate. Alginate alone was used in 9 (20%) children with proved GERD (In table 4 there are 14 (30%) cases). In other patients (n=6, 195 13%) only symptomatic measures were recommended ((In table 4 there are  43 (93%) cases).)

 Specify what the abbreviations mean: SI, SSI, SAP, LLM (lines 126-127, 141)

Respect the journal's requirement for writing references.

Comments on the Quality of English Language

 Minor editing of English language required

Round 2

Reviewer 1 Report

Comments and Suggestions for Authors

The authors have adequately revised

Reviewer 3 Report

Comments and Suggestions for Authors

Accept in present form. I would like to congratulate the authors of the interesting article.

Comments on the Quality of English Language

Minor editing of English language required